# Towards verifying CH<sub>4</sub> emissions from hard coal mines using mobile sun-viewing Fourier transform spectrometry

Andreas Luther<sup>1</sup>, Ralph Kleinschek<sup>7</sup>, Leon Scheidweiler<sup>7</sup>, Sara Defratyka<sup>6</sup>, Mila Stanisavljevic<sup>4</sup>, Andreas Forstmaier<sup>3</sup>, Alexandru Dandocsi<sup>5</sup>, Sebastian Wolff<sup>1</sup>, Darko Dubravica<sup>2</sup>, Norman Wildmann<sup>1</sup>, Julian Kostinek<sup>1</sup>, Patrick Jöckel<sup>1</sup>, Anna-Leah Nickl<sup>1</sup>, Theresa Klausner<sup>1</sup>, Frank Hase<sup>2</sup>, Matthias Frey<sup>2</sup>, Jia Chen<sup>3</sup>, Florian Dietrich<sup>3</sup>, Jarosław Nęcki<sup>4</sup>, Justyna Swolkień<sup>4</sup>, Andreas Fix<sup>1</sup>, Anke Roiger<sup>1</sup>, and André Butz<sup>7</sup>

<sup>1</sup>Deutsches Zentrum für Luft- und Raumfahrt (DLR), Institut für Physik der Atmosphäre, Oberpfaffenhofen, Germany <sup>2</sup>Karlsruhe Institute of Technology (KIT), Institute of Meteorology and Climate Research (IMK-ASF), Karlsruhe, Germany <sup>3</sup>Environmental Sensing and Modeling (ESM), Technische Universität München (TUM), Germany <sup>4</sup>AGH - University of Science and Technology, Kraków, Poland

<sup>5</sup>National Institute of Research and Development for Optoelectronics (INOE2000), Măgurele, Romania

<sup>6</sup>Laboratoire des sciences du climat et de l'environnement (LSCE-IPSL) CEA-CNRS-UVSQ Université Paris Saclay, Gif-sur-Yvette, France

<sup>7</sup>Institut für Umweltphysik, University of Heidelberg, Germany

Correspondence to: Andreas Luther (andreas.luther@dlr.de)

Abstract. Methane (CH<sub>4</sub>) emissions from coal production are one of the primary sources of anthropogenic CH<sub>4</sub> in the atmosphere. Poland is the largest hard coal producer in the European Union with the Polish side of the Upper Silesian Coal Basin (USCB) as the main part of it. Emission estimates for CH<sub>4</sub> from the USCB for individual coal mine ventilation shafts range between 0.03kt/a and 20kt/a, amounting to a basin total of roughly 440kt/a according to the European Pollutant Release and

- 5 Transfer Register (E-PRTR, http://prtr.ec.europa.eu/, 2014). We mounted a ground-based, portable, sun-viewing FTS (Fourier Transform Spectrometer) on a truck for sampling coal mine ventilation plumes by driving cross-sectional stop-and-go patterns at 1 to 3km distance to the exhaust shafts. Using a mass balance approach, several of these transects allowed for estimating CH<sub>4</sub> emissions based on the observed enhancements of the column-averaged dry-air mole fractions of methane (XCH<sub>4</sub>). Our resulting emission estimates range from  $6 \pm 1$  kt/a for a single shaft up to  $109 \pm 33$  kt/a for a subregion of the USCB, which
- 10 is in broad agreement with the E-PRTR reports. Three wind lidars were deployed in the larger USCB region providing ancillary information about spatial and temporal variability of wind and turbulence in the atmospheric boundary-layer. Sensitivity studies show that, despite drawing from the three wind lidars, the uncertainty of the local wind dominates the uncertainty of the emission estimates, by far exceeding errors related to the XCH<sub>4</sub> measurements itself. Wind-related relative errors on the emission estimates typically amount to 20%.

# 15 1 Introduction

Atmospheric methane (CH<sub>4</sub>) causes the second largest radiative forcing of the long-lived greenhouse gases after carbon dioxide (IPCC, 2013). Except for the period from about 1999 to 2006, the global atmospheric CH<sub>4</sub> burden has been rising since 1750

and is now a factor of roughly 2.5 higher than in the pre-industrial era (Nisbet et al., 2014; Miller et al., 2013). Reasons for the renewed rise since 2007 are debated, however. Increased anthropogenic emissions are likely to be part of the answer (Bousquet et al., 2006; Kirschke et al., 2013; Schwietzke et al., 2016; Nisbet et al., 2016; Helmig et al., 2016; Worden et al., 2017).

- Generally, about 20% of the global methane source is thought to be caused by the fossil fuel industry (Schwietzke et al., 2016; Bousquet et al., 2006). The E-PRTR (European Pollutant Release and Transfer Register, http://prtr.ec.europa.eu/, 2014) reports the total European anthropogenic CH<sub>4</sub> emission with 1874kt for the year 2014. Methane emitted by underground mining and related operations amounts to 686kt/a for all reporting facilities in Europe. With 466kt/a the Upper Silesian Coal Basin (USCB) in Poland is a CH<sub>4</sub> emitting hot spot in Europe. The USCB roughly comprises an area of 50km × 50km centered at 50.1N, 18.8E with about 50 active CH<sub>4</sub> ventilating hard coal mining shafts emitting between 0.03kt/a and 20kt/a
- 10 (E-PRTR, 2014). The USCB was the main target of the CoMet (The Carbon dioxide and Methane mission 2018) campaign, which covered roughly three weeks from 23 May to 12 June 2018. During CoMet several aircraft and ground-based instruments were co-deployed to evaluate strategies on how to verify the local  $CH_4$  emissions. Here, we focus on the CoMet measurements of a mobile ground-based, sun-viewing FTS.
- The sun-viewing FTS of the type EM27/SUN developed by the Karlsruhe Institute of Technology in collaboration with Bruker Optics (Gisi et al., 2012) generally deliver the column-averaged dry-air mole fractions of methane (XCH<sub>4</sub>) and other gases by measuring direct-sun absorption spectra in the shortwave-infrared spectral range (around 1650 nm wavelength). Recently, Hase et al. (2015), Frey et al. (2015), and Chen et al. (2016) combined several of these FTS instruments into ad-hoc networks in the vicinity of major cities to estimate urban carbon dioxide (CO<sub>2</sub>) and CH<sub>4</sub> emissions. Viatte et al. (2017) used a similar FTS configuration together with large-eddy simulations to estimate the amount of CH<sub>4</sub> emitted by dairies in southern
- 20 California. Toja-Silva et al. (2017) verified power plant emissions with computational fluid dynamics simulations and differential column measurements using two EM27/SUN in Munich. Klappenbach et al. (2015) demonstrated mobile deployment on a research vessel requiring a custom-built solar tracker to compensate for the motion of the platform. Butz et al. (2017) mounted the EM27/SUN on a small truck to measure the volcanic  $CO_2$  plume of Mt. Etna by recording plume transects in stop-and-go patterns. Kille et al. (2019) separated natural, from agricultural  $CH_4$  emissions using a network of four EM27/SUN. Frey et al.
- 25 (2019) observed an average instrument-to-instrument difference of 0.8 ppb for XCH<sub>4</sub> for an ensemble of instruments. According to the Allan deviations described by Chen et al. (2016) the precision of XCH<sub>4</sub> for 120s integration time is 0.3 ppb, which is roughly 0.02% of the total column.

Based on a mobile FTS setup similar to Butz et al. (2017), here, we explore the feasibility of estimating  $CH_4$  emissions for individual coal mine ventilation shaft and groups of shafts. To this end, we mounted the EM27/SUN on a truck and, we drove

- 30 transects through the  $CH_4$  plumes in the USCB while driving stop-and-go patterns. We estimated emissions from the recorded XCH<sub>4</sub> enhancements using a mass balance method, which was assisted by wind information from three wind lidars deployed in the broader region. Section 2 summarizes the campaign setup, describes the deployed instruments and the data analysis, and explains the emission estimation method. Emission estimates of  $CH_4$  are summarized in section 3 followed by the analysis of the errors in section 4. We evaluate the suitability of the used method, compare the estimated emissions to the E-PRTR
- 35 database, and conclude this work with section 5.

**Figure 1.** Map of the USCB in Southern Poland (see small inset on the left with black illustrating the region of Silesia and red depicting the map excerpt of the USCB). Ventilation shafts are gray triangles. Colored dots depict five plume transects performed by the mobile FTS on 24 May at around 7-8 am (orange), on 24 May around noon (blue), on 1 June (green), on 6 June during morning hours (red), and on 6 June around noon (purple). The most active  $CH_4$  emitters are assumed to be located in the southern part of the USCB, which is why mobile FTS measurements are focused on this area. Stationary EM27/SUN FTS are marked as red triangles; the three wind lidars DLR85, DLR86, and DLR89 are marked as red stars. Eastern and southern wind lidars are placed at the same locations as the respective EM27/SUNs. Background map from ESRI (2019).

### 2 Campaign setup, instruments, and methods

Fig. 1 shows a map of the USCB. The coal mine ventilation shafts group into a northern part in the vicinity of the city of Katowice and into a southern part close to the Czech border. The reported operations of the mobile FTS were carried out in the southern part where the largest emitters are expected. In the framework of CoMet, we also operated three wind lidars and four stationary, sun-viewing FTS of the same model as the mobile one at stations surrounding the USCB. In-situ sensors were deployed in cars sampling throughout the region. Aircraft carrying in-situ (Kostinek et al., 2019) and remote sensing instrumentation (Gerilowski et al., 2011) was operated out of Katowice airport, and a long-distance aircraft (Amediek et al., 2017) visited the USCB operating out of the airport at Oberpfaffenhofen close to Munich, Germany. Here, we use the mobile FTS measurements (section 2.1) and the wind lidar data (section 2.2) together with the cross-sectional flux method (section

10 2.3) to estimate  $CH_4$  emissions. Future studies will target at the other CoMet data and at combining them.

# 2.1 Mobile FTS observatory

5

The mobile FTS observatory consists of a Bruker EM27/SUN Fourier Transform spectrometer with a custom-built solar tracker deployed on a truck similar to the setup used by Butz et al. (2017). The custom-built solar tracker enables efficient stop-and-