# Peer review of "Towards verifying CH4 emissions from hard coal mines using mobile sun-viewing Fourier transform spectrometry"

_Atmospheric Measurement Techniques, 2019_

## Referee Comment (RC1) · Anonymous Referee #1 · 5 Jul 2019

This study presents methane measurements from a mobile solar-viewing IR spectrometer downwind of coal mine vents emitting methane. It includes not only a calculation of plume enhancements, but also emission fluxes. The authors also describe a detailed error budget. Overall I think this is a nice contribution to existing literature, and though I have a lot of comments, they are minor.

Specific Comments

S1: p1l1 - It is unclear if this statement is globally or just for Europe, where it seems about one-third of anthropogenic methane is from coal production. If it's global, please provide a reference in the introduction. In either case, give an approximate percentage.

[Figure]

S2: p1l18 - "2.5 higher" do you mean "2.5 times as high as" or "1.5 times higher"? If pre-industrial is 680 ppb, 2.5 times higher is around 2400 ppb...

S3: p2l16 - This sentence needs to be reworded, it currently sounds like each of the studies of Hase 2015, Frey 2015, and Chen 2016 quantified urban fluxes of $CO_2$ and $CH_4$. However, only Hase quantified $CO_2$, and only Chen quantified $CH_4$. Frey 2015 quantified instrument bias and characterized the ILS in support of Hase 2015.

S4: p2l23-26 - These last 2 sentences are a jump in topic and should be removed from this paragraph. If the authors wish to include this information I suggest it be moved to a paragraph in Sect. 2.1 discussing measurement uncertainties. If the authors also wish to keep the Frey 2015 citation, it could be moved there as well.

S5: p3l2-10 - Please split into 2 paragraphs, one with measurements you use in this study (mobile FTS, wind lidar), one with other ancillary measurements not used here (stationary FTS, aircraft in situ, and aircraft remote sensing). You could also mention anticipating use of all data in a future study.

S6: p4l1 - Quantify "fast"

S7: p4l12 - Please be consistent with wavenumbers or wavelength. Generally I've seen IR measurements reported as wavenumbers, and wavelength for UV-Vis. I'm getting wavelength resolution as around 1.4 nm (but you should check if you go with those units).

S8: p4l14-16: I'm confused by "dwell times" and "observation time." Did you only collect one ten scan average measurement per stop? I thought the 15 "x" symbols on Fig 4a indicated you made 15 stops, and at each stop you made multiple 10-scan averages (grey points), please clarify. Is "dwell time" the total time spent at one stop?

S9: p5l5: Is CalPy used here as well for the FFT?

S10: p5l6: Quantify the accuracy of the retrievals here. Precision is already included elsewhere.

S11: p5l8: More detail is needed on how you define your quality filters here. Is there some cutoff threshold? Maybe a histogram in the supplement of the DC interferogram signal would be helpful?

S12: p5l11 - "other gases" what other gases can be detected? Also I think "quantified" would be a better word that "detected" here - detection by itself is usually not particularly useful.

S13: p5l15 - Is this the first study to scale only the lower part of the a priori profile for EM27/SUN retrievals? Also, some examples of a priori and a posteriori profiles in e.g., the supplement would be helpful.

S14: p5l23 - I understand how you determined background within a single transect, but what are the other background variations you are describing here and how did you observe them?

S15: p5l27 - Please move all information on how measurements were collected to one location (e.g., the paragraph on p4). Does this mean you were at each stop for 20 minutes typically? (If each spectrum takes 120 s).

S16: p5l30 - You have the distance between stops on p10, but it would be helpful to list the approximate distance here as well.

S17: p5l32 - I agree that the $\sim$4 ppb is probably too large for a measure of instrument precision, and most likely is including real variations of XCH4. However, it seems like it would be difficult to model these shorter term variations in XCH4, even with the 3 wind lidars...

S18: p5l34 - "which is not possible [with measurements from] the EM27/SUN". This is not entirely true. See the abstract for "B3.5 A real-time retrieval of greenhouse gases from portable, ground-based Fourier-Transform Spectrometers" here https://iwggms14.physics.utoronto.ca/documents/28/Abstract_Booklet_IWGGMS-14.pdf

S19: p7l8 - Given that dyi is not infinitesimally small, it seems like it would make more sense to represent it as Delta yi

S20: p7l11 - Here you convert back from an average dry-air VMR to a column, so why do you even have Eq. 1 in the first place? It seems like it would make sense to just stick with [CH4].

S21: p8l25 - "we selected all cases" - is this a subset of the 5 measurement tracks, or is this why you think the 5 tracks are good, or were there even more tracks and the 5 included here are a subset?

S22: p9Fig4 - I suggest you draw vertical lines separating the plume from the background. If you wanted to make the figures more information rich, you could include wind speed errors on a secondary axis.

S23: p10l3 - It looks like 2 peaks (though if just point 6 were gone, it would look like 1 peak). Do you think you found a "missing" vent by chance?

S24: p11l11 - I noticed there is no discussion about averaging kernels. This is a critical omission. For total column retrievals this could lead to a $\sim$20% bias in the results. It is unclear what the effect would be here on the lower atmosphere only scaling.

S25: p11l9 - Page 10 promised a more detailed analysis here of the 6 vs. 10 discrepancy for 2 transects of a plume from the same source. However, I find this section lacking in such an analysis. Please include a greater discussion on this. Is this due to real temporal variability in emissions? This seems less likely to me. Or is this due to measurement uncertainty? This second possibility seems more likely. Maybe the uncertainty is not any of the measurements themselves, but rather from the difficulty of accounting for variability on shorter (e.g., 1 minute) timescales, and from modeling eddies, etc. I think reporting both of these numbers is very useful to the community as it shows how good reproducibility is.

S26: p12l1 - 0.4 ppb? Compare with page 2...

S27: p13l22 - It seems unlikely for the vented methane to immediately be mixed uniformly throughout the full PBL. So I would like the authors to try another sensitivity test trying mixing to half of the PBL height (this is also somewhat arbitrary, but has been used by others (e.g., Wu et al., 2018 doi: 10.5194/gmd-11-4843-2018)). Start by recalculating the winds for this smaller mixing depth, as well as the effect of averaging kernels (if layers are fine enough).

S28: p14l11 - This is the first time I'm learning about how you distributed emissions. This information should come in an earlier section.

S29: p14l12 - How variable is the methane ventilation (e.g., 5%, by a factor of 2 or more)? Why is it variable?

S30: p14l13 - The personal communication reference should be omitted here since J. Swolkien is a coauthor and their contributions are listed in the "Author contributions" section.

S31: p14l21 - A 40% difference between transects still seems large. I would like to know the mechanism for variable emissions. See also S29.

S32: p15l10/p4l15 - While increasing the scan time decreased the time needed for ten scans, you should also mention that it also decreased your SNR (likely decreased by about a factor of 2).

S33: p15l14-15 - Of course even wind measurements onboard the truck would not provide the full picture as you are interested in the time varying winds from the source all the way to the truck. I agree though that it could be useful (e.g., these authors put a lidar in a truck: Clements et al., 2018 doi:10.1175/BAMS-D-17-0230.1)

S34: p15l16 - Quantify the confidence here.

S35: p15l18 - Quantify fast (1-2 hours per shaft?) and accuracy of method here.

S36: Title and p13l31 - The "towards verifying" in the title makes it sound like this is

the first of several steps in regular estimates of methane fluxes. If so, what are the next steps? Providing information to policy makers or to mining companies? Adapting the method to require fewer personnel hours? Decreasing the uncertainties? Repeating measurements over a longer time? If the first part of the title were changed "Towards verifying CH4 emissions from hard coal mines" -> "Quantifying methane emissions from hard coal mines in Poland" these questions become irrelevant.

Technical comments

As the editor mentioned, a proofreading may help improve the clarity. This is a long, and not necessarily exhaustive, list of technical corrections.

T1: p1l7 - "distance to" -> "from"

T2: p1l7 - Move "using a mass balance approach" to the end of sentence

T3: p1l13 - "itself" -> "themselves"

T4: p2l1 - omit "however"

T5: p2l5 - "with" -> "as"

T6: p2l6 - "With emissions of 466 . . ."

T7: p2l12 - "a" -> "a single" (this emphasis lets the reader know to not look for measurements from other instruments)

T8: p2l14 - "deliver" -> "are used to measure"

T9: p2l15 - omit "wavelength" it is implicit

T10: p2l23 - "four EM27/SUN" -> "four EM27/SUN instruments"

T11: p2l27 - omit "here"

T12: p2l33 - "used method" -> "method used"

T13: p3Fig1 - "performed by" -> "measured using"

T14: p2Fig1 - "EM27/SUN FTS" -> "EM27/SUN FTS locations"

T15: p3l2 - "Fig. 1" -> "Figure 1"

T16: p4l5 - omit "in"; "distance to" -> "of"; "source" -> "sources"

T17: p4l6 - "Depending on the wind direction we chose the transects" -> "We chose the transects depending on the wind direction"

T18: p4l15 - "our deployment, we tentatively increased" -> "our deployment on June 6 we increased"

T19: p4l16 - omit "This only concerns data collected on June 6." (See also previous comment)

T20: p4l20 - "the two" -> "the standard two"

T21: p4l20 - "proposed by" -> "developed by"

T22: p5l3 - "tracking was" -> "tracking is"

T23: p5l5 - change to "For the retrieval of XCH4 from the FTS measurements we use the"

T24: p5l16 - "EMAC simulation results from a simulation similar to the simulation described" -> "EMAC results from a simulation similar to the one described"

T25: p5l22 - omit "and"

T26: p5l27 - I'm not sure what you are trying to say with this first sentence that is not already known. It could be safely omitted.

T27: p5l30 - "in 2 km distance" -> "within 2 km of the source."

T28: p6l4 - "Three Doppler wind lidars of the type Leosphere Windcube 200S" -> "Three Leosphere Windcube 200S Dopplar wind lidars" Also, please include a reference describing these wind lidars.

T29: p6l8 - "in a" -> "towards the"

T30: p6l15 - "The 75o" -> "The 75o scans"

T31: p6l18 - "can" -> "is"

T32: p7Fig3 - "EDR smaller than" -> "EDR greater than" ?

T33: p7l2 - "tool for" -> "tool typically used for"

T34: p8l6 - "dyi" -> "yi" and "Equ." -> "Eq."

T35: p8l18 - "Fig." -> "Figure"

T36: p8l24 - "a linear" -> "a linear least squares"

T37: p10l2 - "closest other" -> "next closest"

T38: p10l4 - "amounted" -> "were"

T39: p10l6 - "could finish" -> "finished"

T40: p10l9 - "North" - I believe convention is that north and south should not be capitalized here.

T41: p10l15 - "with" -> "as"

T42: p11Table2 - "1 June" -> "1 June, hence no E-PRTR estimate is reported"

T43: p11l5 - "was from" -> "was also from"

T44: p11l6 - Omit "compared to the morning transects"

T45: p11l5,7 - Directions here should be lower-case. Please fix capitalization throughout.

T46: p11l8 - "which calculates to" -> "from which we calculate"

T47: p11l10 - "error bars" -> "errors"

[Figure]

T48: p11l10 - "of the several" -> "of several"

T49: p11l10 - "along" -> "from terms in"

T50: p12l12 - "once" -> "once we"

T51: p12l17 - "large" -> "larger"

T52: p13l4 - "CH4" -> "XCH4"

T53: p13l16 - "averaging to" -> "averaging as"

T54: p13l29 - "1 June. The latter-day" -> "1 June, which"

T55: p14l7 - "error bars" -> "errors"

T56: p14l17 - "amount to" -> "are"

T57: p15l3 - "Mobile FTS emission estimates are best estimated with" -> "Our best estimate using the mobile FTS dataset is"

T58: p15l5 - "can be" -> "are"

T59: p15l5 - omit "listed with"

T60: p15l5 - "Best estimated emissions amount to" -> "Our best estimate using the mobile FTS data is"

T61: p15l6 - "in about" -> "within about"

T62: p15l7 - "distance to" -> "of"

T63: p15l17 - "a mobile" -> "a modified mobile"

Optional

These are additional comments the authors may completely ignore as they may be beyond the scope of this work.

O1: p4Fig. 2 - I am curious how the instrument is attached to the pad. Is it bolted down?

O2: p10l6 - I'm curious about what the issues were.

O3: p13l30 - If you are interested in extending this paper to make it more relevant to studies without wind lidars, you could try including some other estimates of wind and comparing with fluxes derived using the more accurate lidars. E.g., using surface winds (similar to Chen et al., 2016) or STILT (similar to Wu et al., 2018).

O4: p15l21 - This section shows the amount of effort needed to make these measurements, which has implications for scaling this analysis to e.g., other coal mines and shorter revisit times. Satellite data, while supported by teams, do not require them in the field for intensive campaigns such as this one and can often cover much larger areas across the entire globe even. It would thus be interesting to know if TROPOMI could be used to calculate similar enhancements, though perhaps the footprint size is too large ($\sim$7x7 km2) and likely could only be used to get some aggregate USCB flux estimate. I agree CH4 imaging instruments would be useful including possibly GHGSat, and might even be able to be used in lieu of any ground-based mobile FTS measurements for much better scalability.

---

## Referee Comment (RC2) · Anonymous Referee #2 · 19 Aug 2019

The paper by Luther et al reports the field deployment of an EM27/SUN in a coal mining area of Poland. The purpose of this study is to test the EM27/SUN system's ability to estimate the emissions of CH4 above background from fugitive coal seam emissions. The spectrometer itself has been used in a number of studies of greenhouse gases, using the technique of direct solar spectroscopy. The EM27/SUN has successfully measured CO2 and the target gas, CH4, to measure/monitor their background concentrations. Its inherent precision (that is measurement to measurement is « 1ppb in CH4) is such that a measure of enhanced CH4 from coal seam gas emissions would be a valuable addition to the available techniques to estimate this important GHG.

[Figure]

There has been an extensive review of this paper by another referee, who has picked up on a number of issues with the paper, particularly clarifying statements made on the method, improving the flow of the text, and numerous technical writing corrections. The authors have responded to this in detail and made numerous appropriate corrections. The paper is therefore much improved. In this review comments will be made on a couple of specific issues to do with the technique and assumptions made that are central to the purpose of this work.

It is clear from the method that this paper uses, the cross-sectional flux method, that the potential sources of error include the plume enhancement (directly related to the CH4 measurement), the effective wind speed Ueff, and the cross-plume segment Del y. Of these error sources, the dominate error is the use/derivation of the effective wind speed; this dominates the error budget and limits the precision and accuracy of the method. So the first question is: what do the authors consider to be a useful measurement? The authors state that comparing this with independent data, that is, the European Pollutant Release and Transfer Register, is only a "rough comparison". How will we know if this method is successful; there must be a measure of what success looks like in terms of what would be useful to the community (mines, local govt regulations etc).

The paper by Varlon et al stipulates that this method should not be used in calm conditions, that is, with Ueff < 2 ms-1. In Varons study it is suggested using meteorological databases to estimate Ueff at 10 meters; has there been any attempt to compare the lidar wind data with independent meteorological data? The authors did undertake a sensitivity study, and this might imply that such a comparison with independent wind data is not possible.

In terms of the error introduced from Ueff, there is the question of how accurate the estimation of the wind speed is from the lidar, and secondly, how turbulence in the wind flow leads to inherent variability in the wind speed. There is also the variability in the CH4 sources themselves. The authors build these error sources into the error budget.

The text states that these sources, up to 20% or more, are estimated. How is this estimate actually done? In most cases it appears to be based on the standard deviation of data from the lidar for example, or is there also factors based on the operation of the lidar? Perhaps the question is what control did the authors have over the operation of the lidars in terms of direct analyses? Did the authors do this wind speed determination directly?

---

## Author Response (AR1)

Dear Anonymous Referee #1,

    Thank you very much for this comprehensive review. We appreciate the level of detail and your effort. All the specific, as well as technical comments are useful and help improving this work. We answered all questions and implemented your suggestions. Answers are written in italics, changes regarding the manuscript are written in blue italics.

**Specific Comments**

    S1: p1l1 - It is unclear if this statement is globally or just for Europe, where it seems about one-third of anthropogenic methane is from coal production. If it's global, please provide a reference in the
15 introduction. In either case, give an approximate percentage.

*The sentence is reworded to: Methane (CH$_4$) emissions from coal production amount to roughly one-third of European, anthropogenic CH$_4$ emissions in the atmosphere.*

20     S2: p1l18 - "2.5 higher" do you mean "2.5 times as high as" or "1.5 times higher"? If pre-industrial is 680 ppb, 2.5 times higher is around 2400 ppb...

*Changed "2.5 higher" to: 1.5 times higher*

25     S3: p2l16 - This sentence needs to be reworded, it currently sounds like each of the studies of Hase 2015, Frey 2015, and Chen 2016 quantified urban fluxes of CO2 and CH4. However, only Hase quantified CO2, and only Chen quantified CH4. Frey 2015 quantified instrument bias and characterized the ILS in support of Hase 2015.

30 *The sentence is reworded to: Recently, Hase 2015 and Chen 2016 combined several of these FTS instruments into ad-hoc networks in the vicinity of major cities to estimate urban carbon dioxide (CO$_2$) and CH$_4$ emissions, respectively.*

    S4: p2l23-26 - These last 2 sentences are a jump in topic and should be removed from this paragraph. If
35 the authors wish to include this information I suggest it be moved to a paragraph in Sect. 2.1 discussing measurement uncertainties. If the authors also wish to keep the Frey 2015 citation, it could be moved there as well.

*The sentences are moved to Sect. 2.1*

    S5: p3l2-10 - Please split into 2 paragraphs, one with measurements you use in this study (mobile FTS, wind lidar), one with other ancillary measurements not used here (stationary FTS, aircraft in situ, and

aircraft remote sensing). You could also mention anticipating use of all data in a future study.

*The paragraph is split into 2 paragraphs*

S6: p4l1 - Quantify "fast"

*The EM27/SUN internal feedback loop has a few tens of milliseconds. A typical start up phase takes a few seconds, depending on the position of the mirrors relative to the sun.*
*"Fast" is now quantified as: ... it supports start-up and repointing within a few seconds once the solar-tracking is interrupted.*

S7: p4l12 - Please be consistent with wavenumbers or wavelength. Generally I've seen IR measurements reported as wavenumbers, and wavelength for UV-Vis. I'm getting wavelength resolution as around 1.4 nm (but you should check if you go with those units).

*Spectral range is changed to wavenumbers ... and operates in the spectral range $4000$ to $11000\,cm^{-1}$.*

S8: p4l14-16: I'm confused by "dwell times" and "observation time." Did you only collect one ten scan average measurement per stop? I thought the 15 "x" symbols on Fig 4a indicated you made 15 stops, and at each stop you made multiple 10-scan averages (grey points), please clarify. Is "dwell time" the total time spent at one stop?

*The duration for one double-sided interferogram is $12\,s$. At every stop we took about 10 of these observations resulting in an average total integration time of about $120\,s$ per stop. Repointing of the solar tracker und manually starting the measurements took on average another $30\,s$ resulting in typical dwell times of about $2\,min\,30\,s$.*
*In the manuscript, "... $120\,s$ total integration time per observation" is changed to $120\,s$ total integration time per stop.*

S9: p5l5: Is CalPy used here as well for the FFT?

*Yes, we used CalPy for the FFT and for quality filtering.*

S10: p5l6: Quantify the accuracy of the retrievals here. Precision is already included elsewhere.

*We performed several side-by-side measurements with different EM27/SUNs and also the Karlsruhe TCCON station during the preparations for this measurement campaign. The accuracy towards TCCON is based on roughly $3$ hours of side-by-side measurements of the mobile EM27/SUN and the Karlsruhe TCCON station in spring 2017. The calibration factor of $0.9974$ corresponds to an average deviation of $4.7\,ppb$ which is corrected for. Since this work is focused on emission estimation we only consider the difference between plume and background $XCH_4$ which is independent of the total column accuracy.*
*Following sentence is added to the manuscript: The accuracy amounts to roughly $4.7\,ppb$ based on $3$*

*hours side-by-side measurements with the Karlsruhe TCCON station in spring 2017.*

S11: p5l8: More detail is needed on how you define your quality filters here. Is there some cutoff threshold? Maybe a histogram in the supplement of the DC interferogram signal would be helpful?

*We only recorded spectra when the car and instrument stood still and when there was direct sunlight without any clouds. In CalPy we set the DC-threshold to $1\%$ for discarding interferograms. For mobile use, e.g. on a ship, Klappenbach (2015) uses a DC-threshold of $5\%$. During all five transects only one recorded interferogram is discarded due to exceeding the DC-fluctuation threshold.*
*Following sentence is added to the manuscript: We set the DC fluctuation threshold to $1\%$ for discarding corrupt interferograms.*

S12: p5l11 - "other gases" what other gases can be detected? Also I think "quantified" would be a better word that "detected" here - detection by itself is usually not particularly useful.

*We agree that detection is the wrong word here. Quantification, however, is misleading as the reader could think of emission quantification which is not tested. We would therefore change detection to measurements.*
*The sentence is changed to: The spectral bandwidth of the EM27/SUN allows for measurements of $H_2O$, $O_2$, $CO_2$, $CH_4$, CO, (CO in combination with an additional channel described by Hase et al. (2016)) and other gases, e.g. HF and HCl, if present in significant amounts (Butz et al., 2017).*

S13: p5l15 - Is this the first study to scale only the lower part of the a priori profile for EM27/SUN retrievals? Also, some examples of a priori and a posteriori profiles in e.g., the supplement would be helpful.

*Butz et al. (2017) scaled the lower tropospheric part of the vertical profile only as they expected the plume of the volcano Etna to be homogeneously distributed in a certain layer between $3.2$ and $4.9\,km$. This is somewhat similar to our cases as we expect the plume to be homogeneously distributed inside the PBL. We added a $CH_4$ profile example figure in the supplement (Figure 1).*
*Following sentence is added to the manuscript: According to Butz et al. (2017) who only scaled the relevant plume layers of mount Etna in Sicily, we only scaled the relevant layers of the $CH_4$ plumes.*

S14: p5l23 - I understand how you determined background within a single transect, but what are the other background variations you are describing here and how did you observe them?

*We observed linear trends in $CH_4$ concentration measurements when comparing before and after plume measurements. These trends are visible in figure 4a, 4b, 4d. We attributed these trends to the various $CH_4$ sources in the USCB. The stationary EM27/SUN instruments measured strong $CH_4$ gradients within the USCB. To avoid confusion we remove this sentence.*

S15: p5l27 - Please move all information on how measurements were collected to one location (e.g., the paragraph on p4). Does this mean you were at each stop for $20$ minutes typically? (If each spectrum

[Figure]

**Figure 1.** A-priori versus retrieved CH$_4$ profile. Note, that the retrieval only scales the lower part of the a-priori profile up to the the expected maximum PBL-height of roughly $800\,hPa$ ($1700\,m$ above ground). The gray line represents an intra plume measurement.

takes $120\,s$).

*This information is already part of section 2.1 and therefore is removed here. The next sentence is adjusted: The relative standard deviations of all measurements recorded at every stop range ...*

    S16: p5l30 - You have the distance between stops on p10, but it would be helpful to list the approximate distance here as well.

*Information on distance between stops is added: The relative standard deviations of all measurements*
10  *recorded at every stop range from $0.12\,\%$ (roughly $2\,ppb$) on 6 June, when most observations were taken far ($> 40\,km$, stops every $500\,m$) from any source to $0.26\,\%$ (roughly $4\,ppb$) on 24 May, when we sampled the plume in $2\,km$ distance, stopping approximately every $70\,m$.*

    S17: p5l32 - I agree that the $4\,ppb$ is probably too large for a measure of instrument precision, and most
15  likely is including real variations of XCH$_4$. However, it seems like it would be difficult to model these shorter term variations in XCH$_4$, even with the $3$ wind lidars...

*Yes, we agree that the $4\,ppb$ includes atmospheric variability. Indeed, due to the turbulent nature of the plume, this variability cannot be caught by modeling. So, we decided to make it part of the error budget*

*of the method.*

S18: p5l34 - "which is not possible [with measurements from] the EM27/SUN". This is not entirely true. See the abstract for "B3.5 A real-time retrieval of green- house gases from portable, ground-based Fourier-Transform Spectrometers" here https://iwggms14.physics.utoronto.ca/documents/28/Abstract_Booklet_IWGGMS-14.pdf

*Thank you for bringing this to my attention. I did not know this retrieval. It needs to be tested. This part is now omitted in the manuscript: ... to discriminate background from plume enhancements of $CH_4$ in real-time, which is not possible with the EM27/SUN.*

S19: p7l8 - Given that dyi is not infinitesimally small, it seems like it would make more sense to represent it as Delta yi

*Changed to: $\Delta y_i$*

S20: p7l11 - Here you convert back from an average dry-air VMR to a column, so why do you even have Eq. 1 in the first place? It seems like it would make sense to just stick with [CH4].

*$XCH_4$ is required for the background removal to get $\Delta XCH_4$ which is then used to calculate $\Delta \Omega$.*

S21: p8l25 - "we selected all cases" - is this a subset of the 5 measurement tracks, or is this why you think the 5 tracks are good, or were there even more tracks and the 5 included here are a subset?

*In total we performed $10$ transects during the campaign period. We had to cancel two due to upcoming clouds. Three transects were finished without proper sampling of the background before and after the plume due to the lack of on the fly $XCH_4$ measurements.*

S22: p9Fig4 - I suggest you draw vertical lines separating the plume from the background. If you wanted to make the figures more information rich, you could include wind speed errors on a secondary axis.

*The figures are changed according to your suggestions:*

[Figure]

[Figure]

[Figure]

[Figure]

[Figure]

S23: p10l3 - It looks like 2 peaks (though if just point 6 were gone, it would look like 1 peak). Do you think you found a "missing" vent by chance?

*We observed the easternmost of the USCB mines. There are no more known shafts upwind. The peaks are*
5 *likely influenced by atmospheric variability.*

S24: p11l11 - I noticed there is no discussion about averaging kernels. This is a critical omission. For total column retrievals this could lead to a 20% bias in the results. It is unclear what the effect would be here on the lower atmosphere only scaling.

*This work is focused on the emission estimates and therefore we use the difference between background and plume XCH₄ only. This difference probably depends marginally on the averaging kernel. But, we also report XCH₄ values which of course should be correct. With respect to S27 we performed two sensitivity studies. We retrieved the spectra again and scaled once to half of the PBL height, and once, we let the retrieval scale the full atmosphere. The mean difference between retrieved half PBL and full PBL XCH₄ values is $0.6\,ppb$ for plume as well as background measurements. The mean difference between retrieved full atmosphere and full PBL XCH₄ values is $5\,ppb$. This bias is within the error budget for the FTS observations and therefore we omit the discussion about averaging kernels.*

S25: p11l9 - Page 10 promised a more detailed analysis here of the 6 vs. 10 discrepancy for 2 transects of a plume from the same source. However, I find this section lacking in such an analysis. Please include a greater discussion on this. Is this due to real temporal variability in emissions? This seems less likely to me. Or is this due to measurement uncertainty? This second possibility seems more likely. Maybe the uncertainty is not any of the measurements themselves, but rather from the difficulty of accounting for variability on shorter (e.g., 1 minute) timescales, and from modeling eddies, etc. I think reporting both of these numbers is very useful to the community as it shows how good reproducibility is.

*The more detailed analysis is provided in section 5. The reference to section 4 is wrong and therefore adapted.*

S26: p12l1 - 0.4 ppb? Compare with page 2..

*Changed to: $0.3\,ppb$*

S27: p13l22 - It seems unlikely for the vented methane to immediately be mixed uniformly throughout the full PBL. So I would like the authors to try another sensitivity test trying mixing to half of the PBL height (this is also somewhat arbitrary, but has been used by others (e.g., Wu et al., 2018 doi: 10.5194/gmd-11-4843-2018)). Start by re-calculating the winds for this smaller mixing depth, as well as the effect of averaging kernels (if layers are fine enough).

*According to the sensitivity tests discussed in S24, a mean difference of $0.6\,ppb$ between half and full PBL retrievals, results in an average difference of $0.76\,\%$ for the estimated emissions. Averaging wind information for half of the PBL results in average differences of $8\,\%$ of the emission estimates. This is within the wind related error range of up to $20\,\%$.*

S28: p14l11 - This is the first time I'm learning about how you distributed emissions. This information should come in an earlier section.

*Following sentence is added to the caption of Table 2: Bold numbers represent estimated emissions and errors together with the respective E-PRTR 2014 entries in the fifth column , which are the mine-wise reported values distributed evenly to every single listed shaft of each mine.*

S29: p14l12 - How variable is the methane ventilation (e.g., 5%, by a factor of 2 or more)? Why is it variable?

*The data we got from one individual mine suggests, that the concentration in the mine can at least double within one hour. However, we have no information on how accurate these measurements are. The methane concentration in the mine e.g. rises if a new coalbed is opened. Mining deeper (as done in parts of the USCB) also comes along with higher methane emissions as coal from deeper levels generally contains more methane.*

S30: p14l13 - The personal communication reference should be omitted here since J. Swolkien is a coauthor and their contributions are listed in the "Author contributions" section.

*Is omitted.*

S31: p14l21 - A 40% difference between transects still seems large. I would like to know the mechanism for variable emissions. See also S29.

*Please see S29.*

S32: p15l10/p4l15 - While increasing the scan time decreased the time needed for ten scans, you should also mention that it also decreased your SNR (likely decreased by about a factor of 2).

*Text on p4\15 changed to: For the last part of our deployment, we tentatively increased the sampling rate to 40kHz, which resulted in average dwell times of 60s , but also decreased the signal to noise ratio.*

*Text on p15\10 changed to: Enhancing the sampling frequency of the FTS decreased the dwell times significantly (only affects 6 June) but also decreased the signal to noise ratio. However, the relative standard deviation due to averaging of the FTS observations is with 2 ppb small compared to the plume enhancements.*

S33: p15l14-15 - Of course even wind measurements onboard the truck would not provide the full picture as you are interested in the time varying winds from the source all the way to the truck. I agree though that it could be useful (e.g., these authors put a lidar in a truck: Clements et al., 2018 doi:10.1175/BAMS-D-17-0230.1)

*Thank you for this interesting article. We agree that the wind history would also be important to know/-model.*

S34: p15l16 - Quantify the confidence here.

*Confidence added: Summarized, our approach enables the emission estimation of CH$_4$ with good confidence (15 to 30 %).*

S35: p15l18 - Quantify fast (1-2 hours per shaft?) and accuracy of method here.

*Added the quantities: fast = (1 to 1.5 h), accurate = (combined relative error of 15 to 30 %)*

S36: Title and p13l31 - The "towards verifying" in the title makes it sound like this is the first of several steps in regular estimates of methane fluxes. If so, what are the next steps? Providing information to policy makers or to mining companies? Adapting the method to require fewer personnel hours? Decreasing the uncertainties? Repeating measurements over a longer time? If the first part of the title were changed "Towards verifying CH4 emissions from hard coal mines" → "Quantifying methane emissions from hard coal mines in Poland" these questions become irrelevant.

*Title is changed to: Quantifying CH$_4$ emissions from hard coal mines using mobile sun-viewing Fourier transform spectrometry*

**Technical comments**

T1: p1l7 - "distance to" → "from"
*Done!*

T2: p1l7 - Move "using a mass balance approach" to the end of sentence
*Done!*

T3: p1l13 - "itself" → "themselves"
*Done!*

T4: p2l1 - omit "however"
*Done!*

T5: p2l5 - "with" → "as"
*Done!*

T6: p2l6 - "With emissions of 466..."
*Done!*

T7: p2l12 - "a" → "a single" (this emphasis lets the reader know to not look for measurements from other instruments)
*Done!*

T8: p2l14 - "deliver" → "are used to measure"
*Done!*

T9: p2l15 - omit "wavelength" it is implicit
*Done!*

T10: p2l23 - "four EM27/SUN" → "four EM27/SUN instruments"
*Done!*

T11: p2l27 - omit "here"
*Done!*

T12: p2l33 - "used method" → "method used"
*Done!*

T13: p3Fig1 - "performed by" → "measured using"
*Done!*

T14: p2Fig1 - "EM27/SUN FTS" → "EM27/SUN FTS locations"
*Done!*

T15: p3l2 - "Fig. 1" → "Figure 1"
*Done!*

T16: p4l5 - omit "in"; "distance to" → "of"; "source" → "sources"
*Done!*

T17: p4l6 - "Depending on the wind direction we chose the transects" → "We chose the transects depending on the wind direction"
*Done!*

T18: p4l15 - "our deployment, we tentatively increased" → "our deployment on June 6 we increased"
*Done!*

T19: p4l16 - omit "This only concerns data collected on June 6." (See also previous comment)
*Done!*

T20: p4l20 - "the two" → "the standard two"
*Done!*

T21: p4l20 - "proposed by" → "developed by"
*Done!*

T22: p5l3 - "tracking was" → "tracking is"
5  *Done!*

T23: p5l5 - change to "For the retrieval of XCH4 from the FTS measurements we use the"
*Done!*

10  T24: p5l16 - "EMAC simulation results from a simulation similar to the simulation described" → "EMAC results from a simulation similar to the one described"

*I would prefer to keep it that way, since EMAC is short for "ECHAM/MESSy Atmospheric Chemistry" and then the simulation would be missing.*

T25: p5l22 - omit "and"
*Done!*

T26: p5l27 - I'm not sure what you are trying to say with this first sentence that is not already known.
20  It could be safely omitted.
*Done!*

T27: p5l30 - "in 2 km distance" → "within 2 km of the source."
*Done!*

T28: p6l4 - "Three Doppler wind lidars of the type Leosphere Windcube 200S" → "Three Leosphere Windcube 200S Dopplar wind lidars" Also, please include a refer- ence describing these wind lidars.
*Done and reference added!*

30  T29: p6l8 - "in a" → "towards the"
*Done!*

T30: p6l15 - "The 75o" → "The 75o scans"
*Done!*

T31: p6l18 - "can" → "is"
*Done!*

T32: p7Fig3 - "EDR smaller than" → "EDR greater than" ?
40  *Done!*

T33: p7l2 - "tool for" → "tool typically used for"
*Done!*

T34: p8l6 - "dyi" → "yi" and "Equ." → "Eq."
*Done!*

T35: p8l18 - "Fig." → "Figure"
*Done!*

T36: p8l24 - "a linear" → "a linear least squares"
*Done!*

T37: p10l2 - "closest other" → "next closest"
*Done!*

T38: p10l4 - "amounted" → "were"
*Done!*

T39: p10l6 - "could finish" → "finished"
*Done!*

T40: p10l9 - "North" - I believe convention is that north and south should not be capitalized here.
*Done!*

T41: p10l15 - "with" → "as"
*Done!*

T42: p11Table2 - "1 June" → "1 June, hence no E-PRTR estimate is reported"
*Done!*

T43: p11l5 - "was from" → "was also from"
*Done!*

T44: p11l6 - Omit "compared to the morning transects"
*Done!*

T45: p11l5,7 - Directions here should be lower-case. Please fix capitalization throughout.
*Done!*

T46: p11l8 - "which calculates to" → "from which we calculate"
*Done!*

T47: p11l10 - "error bars" → "errors"
*Done!*

T48: p11l10 - "of the several" → "of several"
 *Done!*

T49: p11l10 - "along" → "from terms in"
*Done!*

 T50: p12l12 - "once" → "once we"
*Done!*

T51: p12l17 - "large" → "larger"
*Done!*

T52: p13l4 - "CH4" → "XCH4"
*Done!*

T53: p13l16 - "averaging to" → "averaging as"
 *Done!*

T54: p13l29 - "1 June. The latter-day" → "1 June, which"
*Done!*

 T55: p14l7 - "error bars" → "errors"
*Done!*

T56: p14l17 - "amount to" → "are"
*Done!*

T57: p15l3 - "Mobile FTS emission estimates are best estimated with" → "Our best estimate using the mobile FTS dataset is"
*Done!*

 T58: p15l5 - "can be" → "are"
*Done!*

T59: p15l5 - omit "listed with"
*Done!*

T60: p15l5 - "Best estimated emissions amount to" → "Our best estimate using the mobile FTS data is"

*Done!*

T61: p15l6 - "in about" → "within about"
*Done!*

T62: p15l7 - "distance to" → "of"
*Done!*

T63: p15l17 - "a mobile" → "a modified mobile"
*Done!*

**Optional**
These are additional comments the authors may completely ignore as they may be beyond the scope of this work.

O1: p4Fig. 2 - I am curious how the instrument is attached to the pad. Is it bolted down?

*Yes, it is bolted down at three existing threads at the bottom side of the instrument.*

O2: p10l6 - I'm curious about what the issues were.

*We also tried to use small wind-sondes to measure wind speed and direction at the location of the mobile FTS measurements (which failed due to technical issues). We interrupted the FTS measurements to set up and launch the wind-sonde.*

O3: p13l30 - If you are interested in extending this paper to make it more relevant to studies without wind lidars, you could try including some other estimates of wind and comparing with fluxes derived using the more accurate lidars. E.g., using surface winds (similar to Chen et al., 2016) or STILT (similar to Wu et al., 2018).

*This is planned for a future work.*

O4: p15l21 - This section shows the amount of effort needed to make these measure- ments, which has implications for scaling this analysis to e.g., other coal mines and shorter revisit times. Satellite data, while supported by teams, do not require them in the field for intensive campaigns such as this one and can often cover much larger areas across the entire globe even. It would thus be interesting to know if TROPOMI could be used to calculate similar enhancements, though perhaps the footprint size is too large ($7x7$ km$^2$) and likely could only be used to get some aggregate USCB flux estimate. I agree CH4 imaging instruments would be useful including possibly GHGSat, and might even be able to be used in lieu of any ground-based mobile FTS measurements for much better scalability.

*TROPOMI can detect the CH$_4$ outflow of the whole USCB. In combination with emission models, this can - hopefully - be compared to the stationary EM27/SUN network we also deployed in the area. We compared the stationary FTS data with the sparse TROPOMI XCH$_4$ measurements for the campaign period. However, the data is not mature enough for publication.*
Thank you very much for this review. We answered your questions in all conscience. Answers are written in italics.

**Referee comments**

**#1**

It is clear from the method that this paper uses, the cross-sectional flux method, that the potential sources of error include the plume enhancement (directly related to the CH$_4$ measurement), the effective wind speed U$_{eff}$, and the cross-plume segment $\Delta$y. Of these error sources, the dominate error is the use/derivation of the effective wind speed; this dominates the error budget and limits the precision and accuracy of the method. So the first question is: what do the authors consider to be a useful measurement? The authors state that comparing this with independent data, that is, the European Pollutant Release and Transfer Register, is only a "rough comparison". How will we know if this method is successful; there must be a measure of what success looks like in terms of what would be useful to the community (mines, local govt regulations etc).

*We showed, that it is possible to estimate coal mine CH$_4$ emissions with a mobile FTS in combination with detailed wind information. We also report an error range for the estimates ($15\%$ to $30\%$) which includes uncertainties arising from the method used. We compare our estimates with annual mean values reported by the EPRT-R database for whole mines. We observed single shafts of these mines (the mines in this study operated between $2$ and $4$ shafts) for a short period of time. We call the comparison "rough" as only continuous measurements with multiple repeats during all seasons would help to better estimate an annual average. The problem is, that there is no verified and temporal high resolved emission data. A comprehensive validation can only be realized if we compare our method to others, e.g. inside-shaft measurements of CH$_4$ flux or tracer release experiments. On an operational basis a mobile FTS can be used to estimate emissions fast and flexible e.g. during a leakage event.*

**#2**

The paper by Varon et al stipulates that this method should not be used in calm conditions, that is, with U$_{eff} < 2\,\mathrm{ms}^{-1}$. In Varons study it is suggested using meteorological databases to estimate U$_{eff}$ at 10 meters; has there been any attempt to compare the lidar wind data with independent meteorological data? The authors did undertake a sensitivity study, and this might imply that such a comparison with independent wind data is not possible.

*Correct wind information is one of the key measures of the method used in this study. We had the chance to deploy three wind lidars covering different parts of the area of interest. In our opinion, wind lidar data averaged to one, PBL-representing value is a better assumption than using the $10\,m$ wind speed as a model basis. The standard deviation of the intra PBL wind speed average does not represent the uncertainty of the instrument/measurement itself, but represents atmospheric variability. If the variability inside the PBL is high, the estimated error rises. Referee #1 asked for a sensitivity study, in which we used just the lower half of the PBL's wind information to generate $U_{eff}$. This resulted in an average difference of $8\%$ between "full-" and "half-PBL" emission estimates, which is within the error budget. We did not compare the wind lidar measurements with conventional wind data. The lowest level of the wind lidar data output we used is about $100\,m$ above ground level.*

**#3**

In terms of the error introduced from $U_{eff}$, there is the question of how accurate the estimation of the wind speed is from the lidar, and secondly, how turbulence in the wind flow leads to inherent variability in the wind speed. There is also the variability in the $CH_4$ sources themselves. The authors build these error sources into the error budget. The text states that these sources, up to 20% or more, are estimated. How is this estimate actually done? In most cases it appears to be based on the standard deviation of data from the lidar for example, or is there also factors based on the operation of the lidar? Perhaps the question is what control did the authors have over the operation of the lidars in terms of direct analyses? Did the authors do this wind speed determination directly?

*The wind lidars were deployed, operated, and the data analysis directly performed by the institute of atmospheric physics (DLR in Oberpfaffenhofen). The retrieval of wind speed and direction from radial wind speed measurements of the VAD scans was performed with filtered sine-wave fitting according to the literature cited in section 2.2. The uncertainty of radial wind speed estimates is at the order of $0.2\,ms^{-1}$ and is incorporated into the error budget. This level of uncertainty is particularly critical for the error budget in low wind speed conditions. During the Perdigão 2017 experiment, this error value is evaluated by comparing wind lidar data with wind mast measurements (http://www.pa.op.dlr.de/PERDIGAO2017/references/Kigle_Master_thesis.pdf and https://doi.org/10.1088%2F1742-6596%2F1037%2F5%2F052006).*
*The combined standard deviations for our emissions estimates range between $15\%$ (24 May, noon) and $30\%$ (1 June). Part of these are the wind-specific errors which include the $0.2\,ms^{-1}$ uncertainty, the standard deviation of the vertical average over the PBL, the standard deviation of the temporal average over the whole transect, and the horizontal average based on the fact, that the distance of the FTS to the closest wind lidar never exceeded $33\,km$ and the error related to this distance generally stays below $10\%$ for wind speed and below $3\%$ for wind direction.*

**Quantifying CH$_4$ emissions from hard coal mines using mobile sun-viewing Fourier transform spectrometry**

Andreas Luther[1], Ralph Kleinschek[7], Leon Scheidweiler[7], Sara Defratyka[6], Mila Stanisavljevic[4], Andreas Forstmaier[3], Alexandru Dandocsi[5], Sebastian Wolff[1], Darko Dubravica[2], Norman Wildmann[1], Julian Kostinek[1], Patrick Jöckel[1], Anna-Leah Nickl[1], Theresa Klausner[1], Frank Hase[2], Matthias Frey[2], Jia Chen[3], Florian Dietrich[3], Jarosław Nęcki[4], Justyna Swolkień[4], Andreas Fix[1], Anke Roiger[1], and André Butz[7]

[1]Deutsches Zentrum für Luft- und Raumfahrt (DLR), Institut für Physik der Atmosphäre, Oberpfaffenhofen, Germany
[2]Karlsruhe Institute of Technology (KIT), Institute of Meteorology and Climate Research (IMK-ASF), Karlsruhe, Germany
[3]Environmental Sensing and Modeling (ESM), Technische Universität München (TUM), Germany
[4]AGH - University of Science and Technology, Kraków, Poland
[5]National Institute of Research and Development for Optoelectronics (INOE2000), Măgurele, Romania
[6]Laboratoire des sciences du climat et de l'environnement (LSCE-IPSL) CEA-CNRS-UVSQ Université Paris Saclay, Gif-sur-Yvette, France
[7]Institut für Umweltphysik, University of Heidelberg, Germany

*Correspondence to:* Andreas Luther (andreas.luther@dlr.de)

**Abstract.** Methane (CH$_4$) emissions from coal production  amount to roughly one-third of European anthropogenic CH$_4$ emissions in the atmosphere. Poland is the largest hard coal producer in the European Union with the Polish side of the Upper Silesian Coal Basin (USCB) as the main part of it. Emission estimates for CH$_4$ from the USCB for individual coal mine ventilation shafts range between $0.03\,$kt/a and $20\,$kt/a, amounting to a basin total of roughly $440\,$kt/a

[revised manuscript text omitted]

Here, we use the mobile FTS measurements (section 2.1) and the wind lidar data (section 2.2) together with the cross-sectional flux method (section 2.3) to estimate $CH_4$ emissions. Future studies will target at the other CoMet data and at combining them.

**2.1   Mobile FTS observatory**

5  The mobile FTS observatory consists of a Bruker EM27/SUN Fourier Transform spectrometer with a custom-built solar tracker deployed on a truck similar to the setup used by Butz et al. (2017). The custom-built solar tracker enables efficient stop-and-go patterns since it supports  start-up and repointing  within a few seconds once the solar-tracking is interrupted. During driving, however, significant high frequency mechanical disturbances occur, which is not yet compensated for by the tracker. The mobile FTS observatory was operated out of its campaign-base at the site Pustelnik in the  south of the

10  USCB between 23 May and 12 June 2018 whenever weather conditions were promising for direct-sun viewing. The mobile instrument followed public roads on approximately perpendicular tracks to the assumed plume  1 to 3 km downwind  of the targeted $CH_4$  sources (Fig. 2).  We chose the transects depending on the wind direction  and aimed for isolated shafts with preferably no other shafts upwind. However, we also examined subregions within the USCB containing several mines and shafts upwind. For the plume transects, it was necessary to sample the $CH_4$

15  background on both sides of the plume in order to reliably remove the $XCH_4$ background and to derive the above-background enhancements, which are used for the emission estimation. In total, we report on five transects. One morning transect and one noon transect for 24 May, one transect on 1 June, and one morning transect and one noon transect on 6 June.

The EM27/SUN used here has a spectral resolution of $0.5\,cm^{-1}$ and operates in the spectral range  4000 to 11000 $cm^{-1}$. Frey et al. (2019) observed an average instrument-to-instrument difference of $0.8\,ppb$ for $XCH_4$ for an ensemble of instruments. According to the Allan deviations described by Chen et al. (2016) the precision of $XCH_4$ for 120 s integration time is $0.3\,ppb$, which is roughly $0.02\%$ of the total column. The accuracy amounts to roughly $4.7\,ppb$ based on a 3 hours side-by-side measurements with the Karlsruhe TCCON station in spring 2017. We recorded double-sided interferograms with a sampling rate of $10\,kHz$ and we typically coadded ten double-sided interferograms resulting in roughly $120\,s$ total integration time per stop. Typical dwell times per stop in our stop-and-go patterns were $2\,min\ 30\,s$. For the last part of our deployment  on June 06 we increased the sampling rate to $40\,kHz$, which resulted in average dwell times of $60\,s$.  As depicted in Fig. 3 the FTS is mounted on a suspension plate. The rubber bearings absorb minor shocks and protect the instrument from major bumps. A $12\,V$ battery powers the whole system. We remotely operated the instrument from inside the driver's cabin, which enabled us to measure alongside busy roads as we did not have to leave the car to start the measurements. The solar tracker is the standard two mirror system  developed by Gisi et al. (2011) supplemented with a two-stage tracking control mechanism. The coarse tracking stage uses a fish-eye-camera that identifies the position of the sun and gears the tracker into the vicinity of the sun. The fine-tracking system is the camera system developed by Gisi et al. (2011), which takes over once the coarse tracking  is successful. Generally, the fine-tracking errors amount to $10\,arc\ s$, i.e., the tracking is precise to within fractions of the apparent diameter of the solar disk.

 For the retrieval of $XCH_4$ from the FTS measurements  we use the software package PROFFIT (Hase et al., 2004), which is in routine use for accurate trace gas retrievals from shortwave-infrared direct-sun absorption spectra (e.g., Gisi et al., 2012; Frey et al., 2015; Klappenbach et al., 2015; Chen et al., 2016; Butz et al., 2017; Frey et al., 2019). Quality filtering, based on the unmodulated (DC) part of the recorded interferograms, removes all measurements that are affected by unstable pointing conditions, e.g., due to intermittent thin clouds disturbing solar-tracking (e.g., Klappenbach et al., 2015). We set the DC fluctuation threshold to $1\%$ for discarding corrupt interferograms. The spectral bandwidth of the EM27/SUN allows for  measurements of $H_2O$, $O_2$, $CO_2$, $CH_4$, CO, (CO in combination with an additional channel described by Hase et al. (2016)) and other gases if present in significant amounts (Butz et al., 2017). Related species for this study are $O_2$ and $CH_4$. The spectral retrieval windows for these target absorbers are 7765 to $8005\,cm^{-1}$ and 5897 to $6145\,cm^{-1}$ for $O_2$ and $CH_4$, respectively. The absorption line parameters are taken from Toon (2017) and Rothman et al. (2009). Assuming that the targeted $CH_4$ plumes mainly reside in the planetary boundary layer (PBL), the PROFFIT retrieval was set up to estimate the total column of methane $[CH_4]$, in units of molecules $m^{-2}$, by only scaling the lower part ($< 1700\,m$ a.g.l) of the a priori vertical profile taken from the EMAC model. According to Butz et al. (2017) who only scaled the relevant plume layers of mount Etna in Sicily, we only scaled the relevant layers of the $CH_4$ plumes. Here, we use EMAC simulation results from a simulation similar to the simulation described by Jöckel et al. (2016). Figure A1 compares $CH_4$ profiles of both, a-priori and retrieved $CH_4$ colmuns which are congruent above $1700\,m$ a.g.l..

For the [O$_2$] retrieval we scaled the full a priori profile. The column-averaged dry-air mole fraction of methane (XCH$_4$) is then calculated through

$$\text{XCH}_4 = \frac{[\text{CH}_4]}{[\text{O}_2]} \times 0.20942 \tag{1}$$

where 0.20942 is the atmospheric O$_2$ mole fraction.

5     It is crucial to remove the XCH$_4$ background reliably  to derive the above-background enhancements $\Delta$XCH$_4$, which are then used to estimate the emissions.  Similar to Butz et al. (2017), we used XCH$_4$ measurements before and after crossing a plume to fit the local background linearly in time. We conducted several sensitivity studies to best estimate the background and to quantify the error associated with background removal (see section 4).

10      On average, we recorded ten spectra per stop and averaged the retrieved XCH$_4$ for every stop location. The  relative standard deviations of all measurements recorded at every stop range from 0.12 % (roughly 2 ppb) on 6 June, when most observations were taken far (>40 km, stops every 500 m) from any source to 0.26 % (roughly 4 ppb) on 24 May, when we sampled the plume  within 2 km  of the source, stopping approximately every 70 m. We take the stop-wise standard deviations as a metric for the measurement error

15 that propagates into the emission estimates, which is conservative compared to the small noise error of individual measurements (0.3 ppb).

    On 6 June, we additionally operated a Picarro CRDS (Cavity Ring-Down Spectroscopy) analyzer for in-situ CO, CO$_2$, CH$_4$, H$_2$O on board the mobile observatory to discriminate background from plume enhancements of CH$_4$ in real-time. The CRDS analyzer provides additional support to identify the horizontal extent of the plume

20 without relying on forecast wind fields.

**2.2   Wind lidars**

Three  Leosphere Windcube 200S Doppler wind lidars (Vasiljević et al., 2016; Wildmann et al., 2018) were deployed during the CoMet measurement campaign. The instruments were distributed to the  east, south, and mid-west of the USCB. The exact locations can be

25 found in Tab. 1 and Fig. 2.

| Instr. | Site | Lat. (°N) | Lon. (°E) | m a.s.l |
|--------|------|-----------|-----------|---------|
| DLR85 | Rybnik (W) | 50.0725 | 18.6298 | 253 |
| DLR86 | The Glade (E) | 50.3292 | 19.4155 | 303 |
| DLR89 | Pustelnik (S) | 49.9326 | 18.7998 | 266 |

**Table 1.** Locations of wind lidars deployed during CoMet. The second column lists the site names with their respective cardinal directions relative to the USCB in brackets.

[Figure]

**Figure 4.** Wind lidar profiles of eddy dissipation rate (EDR) on 6 June at the three locations Rybnik/DLR85 (top), The Glade/DLR86 (middle), Pustelnik/DLR89 (bottom). We performed mobile FTS transects on this day in the morning from 7 to 8 am and around noon (9:30 am to 10:30 am). Dashed lines represent the respective PBL heights. EDR  greater than $10^{-4}\mathrm{m}^2\mathrm{s}^{-3}$ corresponds to PBL values. Note, that the PBL height at station Pustelnik/DLR89 is shallower compared to the other two stations. This is generally observed throughout the measurement campaign.

The measurement sites differ to some extent in the surrounding land surface and vegetation. DLR85 is deployed in the  west of the USCB next to a private airfield mostly used for parachuting activities and flight training schools. The land surface  towards the south-westerly direction is flat. However, the airfield is close to a forest located to the  north-west of the wind-lidar. DLR86 is placed in the  east of the USCB, 62 km from DLR85. The area is surrounded by forest. The
5   southern instrument DLR89 is deployed in the vicinity of the barrier lake Goczalkowickie. The linear distance between the instrument and the bank is roughly 500 m.

The Doppler wind lidars were programmed to perform velocity azimuth display (VAD) scans continuously. To retrieve vertical profiles of not only wind but also turbulence, the VAD scans were performed with two different elevation angles. A series of 24 VAD scans with an elevation angle of 75° are followed by six scans with an elevation angle of 35.3°. The 75°
10   scans were chosen to allow retrievals of mean wind speed and direction profiles with a small cone angle, covering at least the whole boundary layer. Filtered sine-wave fitting (FSWF) according to Smalikho (2003) was used to calculate wind speed and wind direction from line-of-sight velocities. While the 75°-scans  is also used to retrieve turbulence kinetic energy dissipation rate, an elevation angle of 35.3° is necessary to derive turbulent kinetic energy, integral length scale and momentum fluxes according to Smalikho and Banakh (2017). As described in Stephan et al. (2018b), boundary-layer height
15   can be estimated through detection of the height at which the eddy dissipation rate (EDR) drops below a value of $10^{-4}\mathrm{m}^2\mathrm{s}^{-3}$. As an example, Fig. 3 depicts EDR and PBL height as retrieved from 75°-scans of the three wind lidars on 06 June 2018.

The uncertainty of wind speed retrievals with the FSWF-method depends on the signal-to-noise ratio of line-of-sight velocity measurements (Stephan et al., 2018a). We consider an uncertainty of $0.2\ \mathrm{ms}^{-1}$, which is particularly critical under low wind conditions.

**2.3 Cross-sectional flux method**

5   The cross-sectional flux method as discussed by Varon et al. (2018) is an established tool typically used for airborne in-situ observations (White et al., 1976; Mays et al., 2009; Cambaliza et al., 2014, 2015; Conley et al., 2016). It has also been applied to airborne $CH_4$ (partial) column measurements (Krings et al., 2011, 2013; Tratt et al., 2011, 2014; Amediek et al., 2017). Based on the mass balance assumption, the strength $Q$ of a $CH_4$ point source such as a coal mine ventilation shaft can be calculated by integrating the product of $CH_4$ concentration enhancements and wind speed along a plume cross-section. Approximating

10   the integral by a sum over all cross-plume measurements $i$, the source strength $Q$ in units $[kg\,s^{-1}]$ is given by

$$Q = \sum_i \Delta\Omega(x_i, y_i)\, U_{\mathrm{eff}}(x_i, y_i)\, dy_i \Delta\Omega(x_i, y_i)\, U_{\mathrm{eff}}(x_i, y_i)\, \Delta y_i \tag{2}$$

where we assume the horizontal coordinates $x_i$ and $y_i$ along and across plume direction, respectively. $\Delta XCH_4$ [ppm] is translated into the plume enhancement $\Delta\Omega$ [$\mathrm{kg\,m^{-2}}$] by

$$\Delta\Omega = \Delta XCH_4 \frac{[O_2]}{0.20942\, N_a} 10^3\, M_{CH_4} \tag{3}$$

15   with the molar mass $M_{CH_4}$ and the Avogadro constant $N_a$. $U_{\mathrm{eff}}(x_i, y_i)$ is an effective wind speed representative of plume dispersion. The distance $dy_i$ $\Delta y_i$ is the cross-plume segment for which the $\Delta\Omega(x_i, y_i)$ enhancement is assumed to be representative. We choose the measurements $i$ to be individual stops of the stop-and-go patterns i.e., we average quantities over individual stops. For estimating wind speed and wind direction, we choose the distance-weighted average of all three wind lidar profiles interpolated to the time of observation, as the baseline wind scenario. The vertical profiles of wind speed (wind

20   direction) are then averaged over the PBL to represent $U_{\mathrm{eff}}$ ($dy_i$) in Equ $y_i$) in Eq. (2).

There are several caveats and sources of error to this method. (1) The measurement vehicle has to follow public roads, which are not always perpendicular to the plume direction. Therefore, we calculate the cross-plume segment $dy_i$ by $\Delta y_i$ by

$$dy \Delta y_i = ds_i\, sin(\alpha_i) \tag{4}$$

where $ds_i$ is the driven distance between two stops of the stop-and-go pattern and $\alpha_i$ is the relative angle between the PBL-

25   averaged wind direction and the driving direction. (2) The calculation of $\Delta XCH_4$ requires a well-defined $XCH_4$ background removal, which translates into the operational requirement to sample background air on both sides of the plume. (3) We assume that the wind direction remains constant over the sampling duration, which typically is 1 to 1.5 hours. (4) The effective wind speed $U_{\mathrm{eff}}$ must be accurate and representative for the observed plume enhancement $\Delta XCH_4$. Relative errors in effective wind speed or its representativeness equal relative errors in estimated emissions, which is particularly striking under low wind speed

30   conditions. We discuss these caveats along with the data and errors in section 3 and 4.

**3   Estimated CH$_4$ emissions**

 Figure 5 displays five transects: one morning and one noon transect on 24 May, one transect on 1 June, and again one morning and one noon transect on 6 June covering different mines and shafts in the USCB. The panels on the left illustrate the locations of observations relative to the nearest mining ventilation shafts. The wind speed and direction is indicated as small wind barbs. Note, that for 1 June the wind speed was below $5\,\mathrm{kn}$ (i.e., always below $2.5\,\mathrm{ms^{-1}}$) and therefore wind barbs are not displayed. However, wind direction measured by the wind lidars reveals a general  south-easterly wind direction. Right-hand side panels show the timeline of all measurements of the EM27/SUN as gray dots together with their respective local stop-wise averages as black crosses. The background XCH$_4$ concentration (dashed black lines in Fig. 5) is based on a linear least squares (in time) fit of the measurements before and after the plume transect. For our analysis, we selected all cases for which (1) we measured a clear background signal before and after crossing the plume, (2) the wind direction was roughly constant as indicated by the wind-lidars, and (3) the overall transect duration does not exceed $1.5$ hours. All transects and their best-estimate emissions calculated with the baseline wind scenario are described in detail in this section and summarized in Table 2. Discussion of errors follows in section 4.

[Figure]

[Figure]

**Figure 5.** XCH$_4$ measured for five transects downwind of the coal mine ventilation in the USCB (panel rows a) to e)) Left panels display the driven path and the locations of each observation. Wind barbs indicate the wind direction. Note, that on 1 June wind speed was very low (<2.5 m s$^{-1}$) and the wind barbs are not resolved. However, the general wind direction was south-east. The targeted mining ventilation shafts are marked as gray triangles. Right hand panels display measured XCH$_4$ as gray dots and respective local stop-wise averages as black crosses. The best-estimate background XCH$_4$ is indicated as dashed black line. Gray areas represent the measurments identified as intra-plume. The black line on top of every right hand side panel display the combined, relative wind error for each measurement. Dashed gray lines in panel a) illustrate all background options contributing to the background related error. Geographic maps from ESRI (2019).

On 24 May in the morning (panels a) in Fig. 5), we drove stop-and-go patterns with stops roughly every 100 m about 2 km west of a ventilation shaft. The target shaft belongs to a mine with four shafts in total. The  next closest ventilation shaft was located about 2 km to the north, and no other known shafts were located upwind. Observed plume enhancements $\Delta$XCH$_4$  were up to 15 ppb with a maximum around 7 am. The best-estimate emissions $Q$ are $6 \pm 1$ kt/a. Later the 5 same day (panels b) in Fig. 5), we observed the same ventilation shaft but driving from south to north. The

transect was interrupted for nearly half an hour due to technical issues from 11:15 to 11:41 UTC, however, we  finished the transect afterwards. Maximum $\Delta XCH_4$ reached roughly 30 ppb around 11:45 am. Estimated emissions $Q$ are $10 \pm 1$ kt/a, nearly 70 % higher than for the morning transect, which is particularly examined in section 5. Throughout our observations in the morning and around noon, the $XCH_4$ background south of the plume is roughly 4 to 5 ppb higher compared to the north.

| Date and time | estimated emissions [kt/a] | combined $\sigma$ | | E-PRTR 2014 [kt/a] | relative standard deviation ($1\sigma$) due to averaging of: | | | | | | | |
|---|---|---|---|---|---|---|---|---|---|---|---|---|
| | | | | | FTS obs. in ppb | | wind speed ($U_{eff}$) in % | | | wind direction ($dy_i$) in % | | |
| | | [kt/a] | % | [kt/a] | XCH4 | backgr. | vert. | horz. | time | vert. | horz. | time |
| 24 May 7 to 8 am | **6** | **1** | 19 | **9.63** | 2 | 0.2 | 13 | 10 | 8 | 2 | 3 | 2 |
| 24 May noon | **10** | **1** | 15 | **9.63** | 4 | 0.3 | 8 | 10 | 3 | 5 | 3 | 3 |
| 01 June 8 to 10 am | **109** | **33** | 30 | - | 3 | 0.6 | 18 | 10 | 9 | 14 | 3 | 12 |
| 06 June 7 to 8 am | **17** | **3** | 16 | **24.3** | 2 | 0.3 | 10 | 10 | 6 | 2 | 3 | 2 |
| 06 June noon | **81** | **13** | 16 | $\sim 80$ | 2 | 0.4 | 8 | 10 | 4 | 4 | 3 | 2 |

**Table 2.** Overview of successful plume transects along with relative standard deviations of the primary sources of uncertainty. Error estimation procedures are explained in the main text. Bold numbers represent estimated emissions and errors together with the respective E-PRTR 2014 entries in the fifth column, which are the mine-wise reported values distributed evenly to every single listed shaft of each mine. Several upwind sources do not allow accurate source attribution on 1 June, hence no E-PRTR estimate is reported. E-PRTR $CH_4$ sources contributing to the observations of the noon transect on 6 June amount to roughly 80 kt/a, if only considering mining ventilation shafts in the near surroundings (20 km radius). However, far upwind (60 km) ventilation shafts can influence the measurements although all mines located in direct wind direction are listed with 0 kt/a and reach 13 kt/a somewhat north of the mean wind direction.

On 1 June (panels c) in Fig. 5), we performed a transect at the western border of the USCB (green dots in Fig. 2). The wind lidar data from the nearby DLR85 (11 km distance to the observations) revealed particularly low wind speeds ($<2.5$ ms$^{-1}$). Source attribution to individual ventilation shafts is challenging since a group of upwind mining shafts contributes to the measured enhancements. $\Delta XCH_4$ values peaked at about 60 ppb around 9:30 UTC at the end of the 1.5 h  north to south transect. The distance of our observatory to the closest mining shaft was always greater than 4 km. The $CH_4$ emission for the group of shafts is estimated  as $109 \pm 33$ kt/a.

Clear sky conditions on the 6 June 2018 enabled us to observe two targets on one day, one in the morning and one around noon. In the morning around 7 am UTC, we started observing a mine with two shafts at the south-east border of the USCB (red dots in Fig. 2). With the wind blowing from east-northeast, we sampled 1 to 2 km west of the shafts (panels d) in Fig. 5). With the CRDS instrument on board, we were able to assess the rough position of the plume as we could see ground-based in-situ enhancements online while moving the truck. Once we reached background $CH_4$ levels in the south, we started sampling with the FTS and moved northward with an average step size of 100 m. Maximum enhancements $\Delta XCH_4$ reached roughly 35 ppb around 7:30 am. Estimated emissions $Q$ amount to $17 \pm 3$ kt/a. We measured a difference of $+4$ ppb between southern and northern background values.

[Figure]

**Figure 6.** Relative differences between the best-estimate (weighted averaged) wind speed and sensitivity calculations using wind speed from individual wind lidars. Different colors represent different transects. Individual symbols refer to the wind information source used. The right panel is a zoom of the left panel, but it omits data from 1 June.

For the second transect on 6 June (purple dots in Fig. 2 and panels e) in Fig. 5), the wind was also from north-easterly directions. We aimed at measuring a cross section through the southern part of the USCB sampling roughly every $500\,\text{m}$ moving from south to north. Downwind (by about 1 to $2\,\text{km}$) of a group of four shafts close to our track, we measured XCH$_4$ enhancements exceeding $30\,\text{ppb}$, from which we calculate total emissions $Q$ of $81 \pm 13\,\text{kt/a}$.

**4   Error analysis**

The errors reported for the emission estimates in section 3 are composed of several contributions from terms in equation (2): the estimated errors for $\Delta\Omega$ which partition into the measurement error and the error for background removal; the errors associated with effective wind speed $U_{\text{eff}}$, which partition into errors related to vertical, horizontal, and temporal averaging of the wind-lidar measurements; and the errors in $\Delta y_i$, which are dominated by the errors in the relative angle between wind directions and driving directions.

As a measure for the measurement error attributed to XCH$_4$, we take the standard deviation among all measurements collected during an individual stop of our stop-and-go pattern. These standard deviations exhibit a maximum of $4\,\text{ppb}$ for the noon transect on 24 May. Short distances to the mining shaft and wind speeds around $6\,\text{ms}^{-1}$ imply large variability in the CH$_4$ column above the FTS. For the same day in the morning, the relative CH$_4$ standard deviations for a transect at the same shaft were smaller ($2\,\text{ppb}$). The actual instrument precision of the FTS XCH$_4$ measurements is on the order of roughly 0.3 ppb ($0.02\,\%$). Thus, the error estimate is driven by atmospheric variability.

The error associated with XCH$_4$ background removal is estimated as the standard deviation of all possible combinations of non-plume signals observed before and after crossing the plume during the generally $1$ to $1.5\,$h long transects. We define an observation as intra-plume if the absolute difference between two consecutive measurements is at least $0.5\,$ppb greater than the average standard deviation of all measurements. We then linearly fit all possible background estimates as illustrated by gray

5     dashed lines in panel a) of Fig. 5. The average of all these background options is defined as the best estimate background. Their standard deviation is considered in the error budget. The best-estimate XCH$_4$ background is then used to obtain the plume enhancements $\Delta$XCH$_4$ by subtracting the background from the measured XCH$_4$. Overall the background removal standard deviation is smaller than $0.6\,$ppb which is small compared to other sources of error. For 6 June, we rely on the onboard in-situ measurements to locate the start and end point of the plume. We observed the in-situ concentrations on the fly and started with

10     the FTS measurements once we observed constantly low CH$_4$ concentrations.

Errors related to the estimate of effective wind speed $U_{\mathrm{eff}}$ typically dominate the error budget. Our baseline wind scenario averages the lidar wind profiles vertically throughout the PBL and considers the errors arising from temporal wind speed variability throughout one plume transect and from distance-weighted averaging between the three wind lidars. We take the standard deviations among all wind speed measurements inside the PBL to quantify the error related to vertical averaging.

15     The respective error estimates range from $8\,\%$ to $18\,\%$, with  larger errors occurring under low wind speed conditions as observed on 1 June ($U_{\mathrm{eff}} < 2.5\,\mathrm{ms}^{-1}$). The error due to temporal averaging of wind speed is estimated by the respective standard deviations of wind speed during the averaging period. This error ranges from $3\,\%$ to $9\,\%$.

Having the advantage of three wind lidars co-deployed in the USCB, we can also assess $U_{\mathrm{eff}}$ errors related to horizontal variability of wind speeds. To this end, we conducted a sensitivity study that calculates the emissions $Q$ with wind information

20     from each single wind lidar instrument, instead of using a horizontal average that is weighted by the distance between lidar and FTS as in our baseline scenario. Distances between XCH$_4$ observations and wind measurements range between $11\,$km and $72\,$km. Figure 6 shows the relative differences of calculated wind speeds $U_{\mathrm{eff}}$ for all sensitivity runs with respect to the best estimate baseline scenario (reported in section 3).

Differences between the wind speeds generally increase the larger the distance between the FTS and the wind lidar locations.

25     Differences range between $1\,\%$ and $13\,\%$ except for distances exceeding $40\,$km and for low wind speed conditions such as encountered on 1 June, which implies larger errors. For 1 June with $U_{\mathrm{eff}} < 2.5\,\mathrm{ms}^{-1}$, relative wind speed differences differ substantially if wind information is taken from measurements more than a few ten kilometers away. Figure 6 also reveals, that using wind information from DLR89 causes emission estimates that are higher than for the other wind information sources. Wind measurements of DLR89 are likely influenced by the nearby lake, especially during the prevailing easterly winds. Low

30     surface roughness and low friction above the lake surface could cause the wind speeds – and therefore the emission estimates – to be higher in lower levels compared to the other sites located close to forests. Since the distance of the FTS to the closest wind lidar never exceeded $33\,$km and since we used distance-weighted averages of the wind measurements, we estimate the error due to horizontal wind speed averaging  as $10\,\%$.

Errors due to wind direction translate into errors of the cross-plume segment $\Delta y_i$. Similar to the error estimation proce-

35     dures for wind speed, we estimated wind direction errors due to vertical, temporal and horizontal averaging. In general, these

[Figure]

**Figure 7.** The wind rose indicates wind direction and wind speed as occurred during the transects. Different colors mark different days. Size of the circles corresponds to the relative standard deviation in % resulting from averaging wind speed horizontally, within the PBL, and over time. Solid circles mark the best-estimate wind information whereas transparent circles mark wind information that stems from the individual wind lidars directly. During the measurement periods, easterly winds prevailed. Note that for 1 June wind speeds are exceptionally low.

errors are smaller compared to wind speed errors. Vertical averaging yields a standard deviation of $2\%$ to $5\%$ except for the low wind speed day 1 June ($14\%$). Temporal averaging induces errors ranging between $2\%$ and $3\%$ again except for 1 June with $12\%$. We estimate the error due to horizontal wind direction averaging with $3\%$ according to the procedure we used to estimate the horizontal wind speed averaging error.

5    Generally, we take the above individual error contributions, and we propagate them into the reported emission errors for $Q$ by Gaussian error propagation. The error contributions from estimating the effective wind-speed $U_{\text{eff}}$ dominate the error budget. An overview of all wind situations for all transects is given in Figure 7. Every circle refers to a certain time during the transect and the circle size indicates the total relative $U_{\text{eff}}$ error (calculated through Gaussian error propagation from the individual contributions given in Table 2). Solid circles mark best-estimate wind information. Transparent circles indicate data from our

10   sensitivity study based on individual wind lidars. Mainly easterly winds were observed for the transects reported in this study. Wind speeds varied between $4\,\text{ms}^{-1}$ and $8\,\text{ms}^{-1}$ for all cases except for 1 June. , which was challenging for emission estimation since wind speeds were generally low and variable in the deployment region.

**5   Discussion & Conclusion**

We demonstrate ground-based remote sensing of XCH$_4$ enhancements in the plumes of hard coal mining ventilation shafts

15   several kilometers downwind of the sources. Our observatory truck was equipped with an EM27/SUN FTS measuring direct-sunlight near-infrared absorption spectra. With transect periods of $1$ to $1.5\,\text{h}$, the plumes were crossed in stop-and-go patterns. In combination with the data of 3 co-deployed wind lidars we were able to estimate the emissions for five transects on three days.

Our error analysis includes errors related to the FTS measurements and errors related to vertical, horizontal, and temporal averaging of wind speed and wind direction. Combined through Gaussian error propagation these errors range between $15\%$ and $30\%$.

Generally, our best-estimate emissions and the respective  errors show broad agreement with the E-PRTR report (European Pollutant Release and Transfer Register, http://prtr.ec.europa.eu/, 2014), which contains emission data reported annually by industrial facilities over Europe including the coal mining branch. This data set provides annual mean values of $CH_4$ emissions for each mining facility registered. The annual means refer to whole mines, which typically have several shafts to ventilate $CH_4$. We distributed the annual means reported by E-PRTR evenly to all known shafts of every mine. However, our measurements are restricted to a few hours per shaft, and ventilating $CH_4$ of coal mines is a variable process depending on the concentrations in the mine. Furthermore, turbulent processes influence the shape and position of the plume and therefore the measurements. Thus, comparing our snapshot-like measurements with the disaggregated E-PRTR reports can only provide a rough comparison.

Tab. 2 lists the estimated and the reported, and disaggregated emissions for all five transects. On 24 May, we observed a single shaft in the  south-east of the USCB. The reported mine emissions  are 9.63 kt/a. The cross-sectional flux method yields $6 \pm 1$ kt/a for the morning transect and $10 \pm 1$ kt/a for the transect around noon. Background $CH_4$ exhibits a  south to north gradient, indicating that the measurements could have suffered from $CH_4$ inflow from unknown sources. An observed low-level jet is likely influencing the morning transect. However, differences between single transects are likely as the ventilation system is not emitting constantly. Source attribution becomes challenging measuring downwind of various mines and groups of shafts. Hence, no specific reported emission can be assumed for the transect on 1 June. However, reported emissions likely range between 50 kt/a (sum of nearest upwind mines $< 10$ km) and 150 kt/a (sum of south-eastern upwind part of the USCB).  Our best estimate using the mobile FTS dataset is $109 \pm 33$ kt/a. Reported (24.3 kt/a) and estimated emissions ($17 \pm 3$ kt/a) generally agree for the morning transect of 6 June. The noon transect on 6 June crossed several possible $CH_4$ sources although the reported emissions  are roughly 80 kt/a.  Our best estimate using the mobile FTS data is $81 \pm 13$ kt/a, which includes the reported value in the error range. However, many sources are located upwind in the northern part of the USCB  within about 60 km distance  of the measurements, which also could have influenced the observations but are not included in the sum of the reported emissions for that day.

Enhancing the sampling frequency of the FTS decreased the dwell times significantly (only affects 6 June). This made faster transects possible and helped to avoid changing wind and emission conditions. Faster tracking of the sun would be necessary to allow measuring while driving. Wind information is crucial when using the cross-sectional flux method as relative changes in wind speed equal relative changes in estimated emissions. Particularly low wind speed situations suffer from large discrepancies in the estimates when using wind information from different instruments. Thus, the error budget is dominated by atmospheric variability. Precise wind measurements on board the observational truck would help to reduce the wind induced errors. $CH_4$ imaging instruments could guide the mobile FTS providing position and extent of the target plume.

Summarized, our approach enables the emission estimation of $CH_4$ with good confidence (15 to 30 %). However, the method is restricted to direct sunlight and stable wind conditions. Together with detailed wind information and in-situ $CH_4$ measurements, a modified mobile FTS is a flexible, fast (1 to 1.5 h), and accurate (combined relative error of 15 to 30 %) possibility to estimate coal mine $CH_4$ emissions reliably.

**6 Data availability**

The data are available from the author upon request.

*Author contributions.* Andreas Luther wrote the paper except for section 2.2 which was written by Norman Wildmann who also took an active part during the campaign deploying the wind lidars and retrieving and providing wind lidar data. Andreas Luther, Ralph Kleinschek, Leon Scheidweiler, Sebastian Wolff, Sara Defratyka, Mila Stanisavljevic, Andreas Forstmaier, Alexandru Dandocsi, and Darko Dubravica operated the EM27/SUN spectrometer in the field during the campaign and collected and shared the data. Ralph Kleinschek and Julian Kostinek developed the suntracker software which made mobile measurements possible. Patrick Jöckel and Anna-Leah Nickl supported the selection of the target shafts with $CH_4$ forecasts for the region. Theresa Klausner porvided shaft-wise E-PRTR geolocation and emission data retrieved from the E-PRTR dataset (E-PRTR 2014). André Butz, Frank Hase, Matthias Frey, Jia Chen, and Florian Dietrich supported preparations for the measurement campaign, contributed to the spectral retrievals, and assisted with data postprocessing. Jarosław Nęcki and Justyna Swolkień provided detailed information about coal mining and $CH_4$ ventilation and functioned as local advisers. Andreas Fix coordinated the CoMet campaign operations. André Butz and Anke Roiger developed the research question.

*Competing interest.* The authors declare that they have no conflict of interest.

**Appendix A**

[Figure]

**Figure A1.** A-priori versus retrieved $CH_4$ profile. Note, that the retrieval only scales the lower part of the a-priori profile up to the the expected maximum PBL-height of roughly 800 hPa ( 1700 m above ground). The gray line represents an intra plume measurement.